# Iberian Margin surface ocean cooling led freshening during Marine Isotope Stage 6 abrupt cooling events

Hongrui Zhang [1] ✉, Yongsong Huang [2], Reto Wijker[1], Isabel Cacho [3], Judit Torner[3], Madeleine Santos[1], Oliver Kost [1], Bingbing Wei[4] & Heather Stoll [1]

The high-resolution paleoclimate records on the Iberian Margin provide an excellent archive to study the mechanism of abrupt climate events. Previous studies on the Iberian Margin proposed that the surface cooling reconstructed by the alkenone-unsaturation index coincided with surface water freshening inferred from an elevated percentage of tetra-unsaturated alkenones, $C_{37:4}$%. However, recent data indicate that marine alkenone producers, coccolithophores, do not produce more $C_{37:4}$ in culture as salinity decreases. Hence, the causes for high $C_{37:4}$ are still unclear. Here we provide detailed alkenone measurements to trace the producers of alkenones in combination with for-aminiferal Mg/Ca and oxygen isotope ratios to trace salinity variations. The results indicate that all alkenones were produced by coccolithophores and the high $C_{37:4}$% reflects decrease in SST instead of freshening. Furthermore, during the millennial climate changes, a surface freshening did not always trigger a cooling, but sometimes happened in the middle of multiple-stage cooling events and likely amplified the temperature decrease.

The high sedimentation rate of the Iberian Margin has made it an essential reference for detecting millennial climatic coolings, which characterized the North Atlantic region during glacial periods[1–5]. In combination with records of abrupt events in Greenland and Antarctic ice cores[6], as well as ice-rafted debris (IRD)[7], percentages of *Neogloboquadrina pachyderma* in the higher latitude North Atlantic[8] and loess grain size in the Chinese Loess Plateau[9], the Iberian margin temperature records elucidate the regional climate response on millennial scales. On centennial to millennial timescales, these abrupt coolings were usually attributed to the superposition of weakened Atlantic Meridional Overturning Circulation (AMOC) on an already cool climate background during glacial[10]. The reduction of AMOC was often attributed to ice melting and surface freshening in the North Atlantic[11, 12], as simulated in numerous "hosing experiments" on coupled ocean-atmosphere models[13–15]. However, at least in some locations, the temperature decrease preceded the arrival of IRD,

suggesting that freshening from collapse of marine-based ice sector may not have been the initial trigger of the AMOC reduction and abrupt cooling[8].

The continuous and high-resolution records from alkenones, organic biomarkers produced by three groups of haptophyte[16], have provided a detailed view of the interplay of millennial and orbital scale cooling events on the Iberian Margin (Fig. 1). In addition to the lower alkenone unsaturated index ($U^{K}_{37}$ = $C_{37:2}/(C_{37:2} + C_{37:3})$), an indicator of colder sea surface temperature (SST), many stadial events featured a higher percentage of tetra-unsaturated alkenone ($C_{37:4}$% = $C_{37:4}/(C_{37:4} + C_{37:3} + C_{37:2})$), which has been variably interpreted as an indicator of the presence of either Arctic-like waters, cold temperatures, or freshening of the surface ocean[2–5]. If the abundance of tetra-unsaturated alkenone were a proxy of freshening, it would provide a direct way to explore the relationship of freshening with abrupt millennial coolings.

[1]Geological Institute, ETH Zürich, 8092 Zürich, Switzerland. [2]Department of Earth, Environmental and Planetary Sciences, Brown University, Providence RI 02912, USA. [3]Grup de Recerca Consolidat en Geociències Marines, Department de Dinàmica de la Terra i de l'Oceà, Universitat de Barcelona, Barcelona, Spain. [4]Alfred Wegener Institute, Helmholtz Center for Polar and Marine Research, Bremerhaven 27570, Germany. ✉e-mail: zhh@ethz.ch

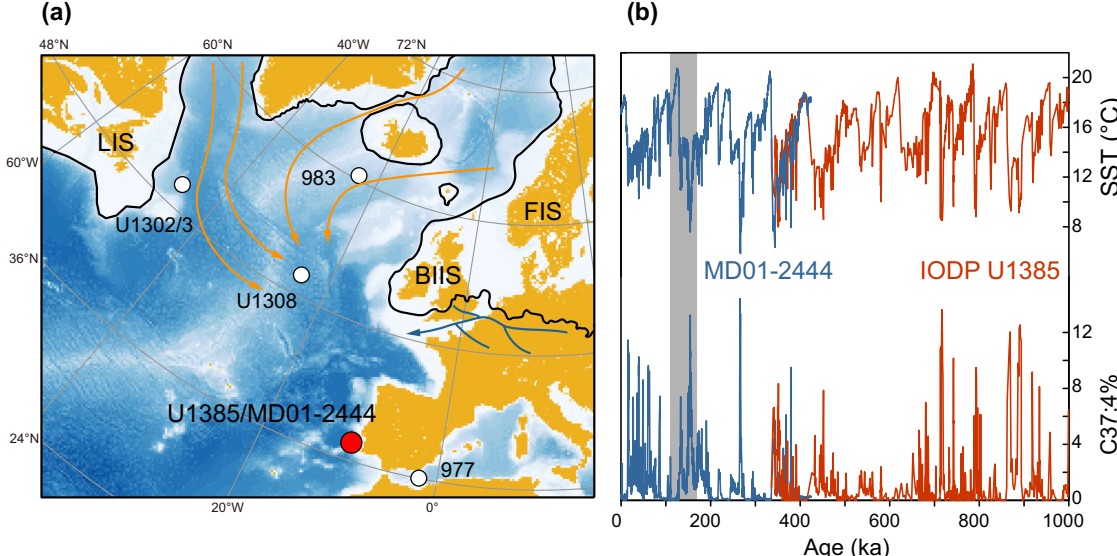

**Fig. 1 | Sites locations and climate records on the Iberian Margin. a** Sites locations. The black lines mark the estimated ice sheet edge during the Marine Isotope Stage (MIS) 6[98]. The orange lines in the ocean are the trajectories of ice rafted debris[40]. The dark blue lines represent the position of Fleuve Manche[99]. The FIS, BIIS, and LIS are the Fennoscandian ice sheet, British Isles ice sheet, and Laurentide ice sheet, respectively. **b** The alkenone-based reconstructions on the Iberian Margin[4,5]. The shaded area represents the period of interest in this work.

The $C_{37:4}$ can be produced by all three groups of alkenone producers. The algae of Group 1 live in freshwater lakes in the northern Hemisphere[17,18]. Group 2 could be found in brackish, saline lakes, costal region, and sea ice[19–21]. Group 3 is mainly be composed by marine algae, coccolithophores, including *Emiliania huxleyi*, *Gephyrocapsa* spp., and their ancestors, *Reticulofenestra* spp.[22–25]. Despite a common interpretation of higher $C_{37:4}\%$ as lower salinity in the oceans[26,27], however, the impact of salinity on $C_{37:4}\%$ is still ambiguous. Positive or no correlation between salinity and $C_{37:4}\%$ can be found in culture experiments of alkenone producers in Group 2 and Group 3[20,28–33]. Negative correlations between salinity and $C_{37:4}\%$ in the surface sediment of the North Atlantic[27,34] and the Baltic Sea[35] are mainly caused by changes in the relative contribution of different groups of alkenone producers, due to either sea ice dynamics[20] or a large local salinity gradient[36]. A recent culture study documented that the low temperature and high growth rate could stimulate the production of $C_{37:4}$ by two Arctic strains of *E. huxleyi*, challenging the classical explanation of $C_{37:4}\%$ as a proxy of salinity.

Consequently, the interpretation of $C_{37:4}\%$ and the usage of alkenone thermometers on the Iberian Margin remain uncertain with multiple possible hypotheses. If the high $C_{37:4}\%$ is explained as low salinity, it must reflect mixings between Group 1 or Group 2 alkenones and Group 3 alkenones. The mixing explanation of $C_{37:4}\%$ would cause a bias in the sea surface temperature reconstructed by the $U_{37}^{K'}$, because the alkenones produced by Group 1 and Group 2 algae may have different $U_{37}^{K'}$ than that produced by Group 3 at the same temperature[37,38]. On the other hand, if the source of alkenone on the Iberian Margin remained stable, as originating from Group 3 producers[39], and no physiological response to salinity has been detected in Group 3, what process is responsible for the episodes of higher $C_{37:4}\%$ on the Iberian Margin during glacial periods?

To better constrain the mechanism of abrupt climate events, we present a detailed study on the alkenones and surface seawater salinity during the late Marine Isotope Stage (MIS) 6 and the onset of deglaciation, a period with high amplitude $C_{37:4}\%$ peaks[4], and extensive Eurasian-sourced IRD[40]. An accurate interpretation of the boundary conditions of late MIS 6 glacial is also relevant to Paleoclimate Modelling Intercomparison Project (PMIP) protocol for the last deglaciation[41]. During the studied period, two significant $C_{37:4}\%$ peaks were detected in multiple sites along the Iberian Margin (Fig. 1b). The one at ~155 ka was named as 'Mid-MIS 6 event' (MME) by Margari et al.[42] and another at ~133 ka was widely attributed to Heinrich event 11 (HE11). In this study, multiple proxies are utilized to better discern the environmental signals on the Iberian Margin. The fingerprints of alkenones ($C_{37}$–$C_{39}$) were carefully examined to identify the producers of alkenones. The alkenone thermometers, $U_{37}^{K'}$, $U_{37}^{K} = (C_{37:2}\text{-}C_{37:4})/(C_{37:2} + C_{37:3} + C_{37:4})$ and $U_{38ME}^{K'} = C_{38:2ME}/(C_{38:2ME} + C_{38:3ME})$, were calculated and compared with published simulations. The abundance of alkenones as well as coccoliths, carbonate fossils of coccolithophores, was quantified to estimate the paleo productivity. The paired Mg/Ca ratio and oxygen isotope ($\delta^{18}O$) of the carbonate shell of foraminifera were analyzed to estimate seawater oxygen isotope ratios, which could indirectly indicate changes in salinity. These results allow us to refine the interpretation of alkenone proxies and provide clearer constraints on the mechanism of these two millennial events.

## Results and discussion
### No significant shifts in alkenone producers
Before looking at the paleo-reconstructions, it is necessary first to distinguish the alkenone producers on the Iberian Margin. In this work, the potential source of alkenones was traced by alkenone assemblages (Fig. 2a) because the $C_{37:4}\%$ and ratios of unsaturated methyl and ethyl alkenones with 38 carbon atoms ($C_{38ME}:C_{38ET}$) from different producers are characterized by distinct values[36,43]. Briefly, the $C_{37:4}\%$ of Group 1 is larger than 50%, the $C_{37:4}\%$ of Group 2 is ~5–50% (Fig. 2a) and the $C_{37:4}\%$ of Group 3 significantly increases in cold water[44]. The algae in Group 1 usually produce alkenones with the $C_{38ME}:C_{38ET}$ ratios below 0.5[19,36]. The haptophytes in Group 2 produce no or very trace amounts of $C_{38ME}$, resulting in $C_{38ME}:C_{38ET}$ ratios around zero[43]. Coccolithophores in Group 3 feature a wider range of $C_{38ME}:C_{38ET}$[45–49] increasing from tropical to high latitude oceans.

The downcore alkenones on the Iberian Margin (red diamonds in Fig. 2a) are characterized by $C_{37:4}\%$ ranging from 0% to 9.9% and $C_{38ME}:C_{38ET}$ varying from 0.5 to 0.8 (typical gas chromatographic results are in Supplementary Fig. S1). If the alkenone on the glacial Iberian Margin were mixtures between Group 1/Group 2 and Group 3, analogous to the range of the modern Baltic Sea, then we should observe a negative correlation between $C_{37:4}\%$ and $C_{38ME}:C_{38ET}$ as

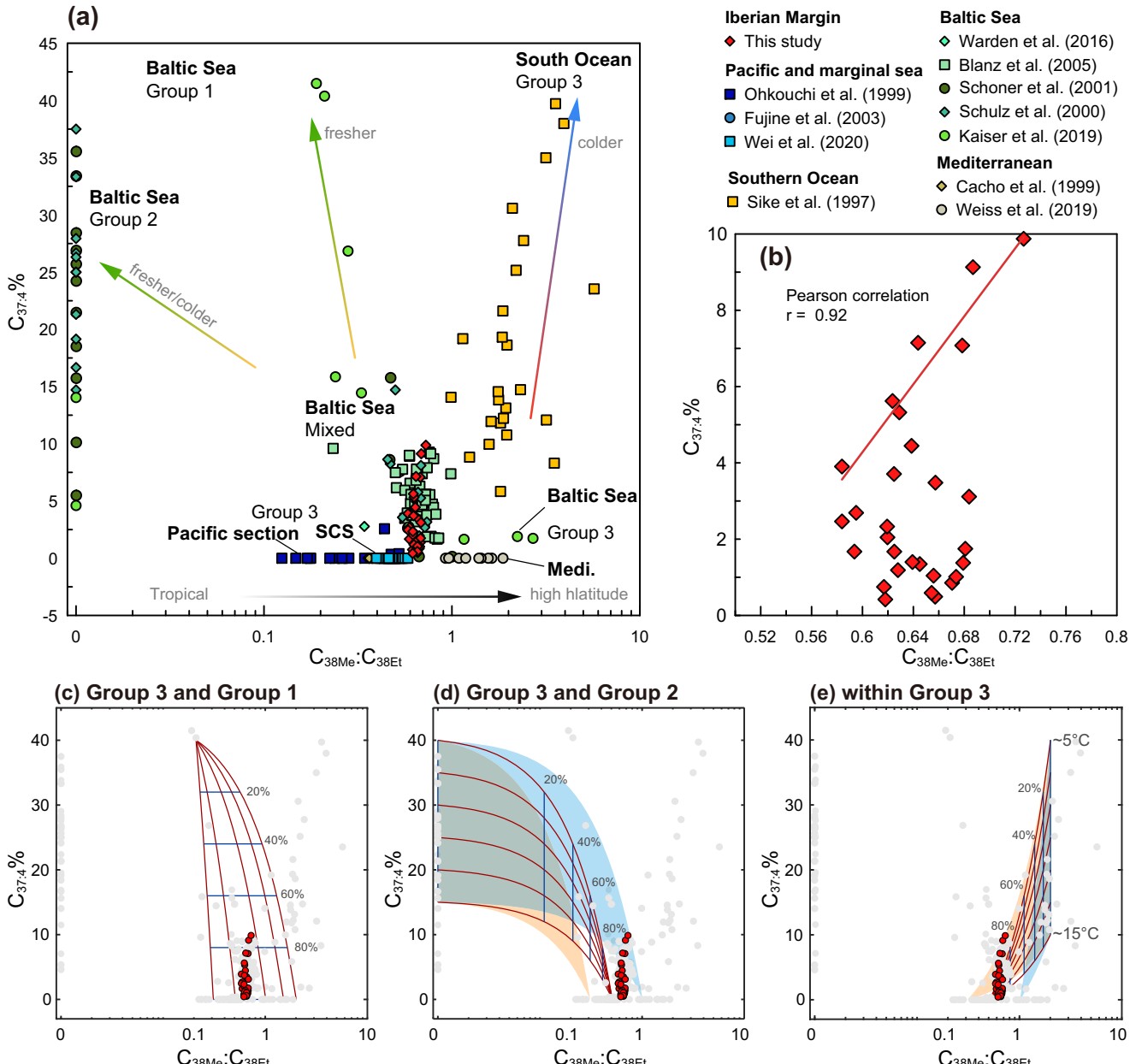

**Fig. 2 | Potential mixing model among different alkenone producers revealed by the cross-plot of ratios of unsaturated methyl and ethyl C$_{38}$-alkenones (C$_{38ME}$:C$_{38ET}$) and tetra-unsaturated alkenones percentage (C$_{37:4}$%). a** Cross-plot of C$_{38ME}$:C$_{38ET}$ and C$_{37:4}$% using published surface sediments datasets[35,36,45-49,100-103] and downcore results in this study (red diamond). Note the x-axis is in log-scale and the data points with C$_{38ME}$:C$_{38ET}$ = 0 were plotted at the position of 0.01 for the convenience of data visualization. The SCS represents the South China Sea and Medi represents the Mediterranean. **b** C$_{38ME}$:C$_{38ET}$ and C$_{37:4}$% of the IODP U1385 samples measured in this work. The C$_{38ME}$:C$_{38ET}$ and C$_{37:4}$% show a significant positive correlation in the sample with C$_{37:4}$% > 4%. **c–e** The potential mixing lines between Group 3 living in tropical-subtropical and Group 1 living in freshwater (**c**), between Group 3 living in tropical-subtropical and Group 2 (**d**), and between Group 3 living in tropical-subtropical and Group 3 living in high latitude cold water (**e**). The blue and orange shaded areas in (**d, e**) represent mixing lines with different end-members of Group 3 C$_{38ME}$:C$_{38ET}$ ratios. Detailed endmember descriptions are listed in Supplementary Note Table S1. Red and gray dots in (**c, d**) represent downcore measurements in this study and global surface sediment in previous works, respectively. Source data of this figure are provided as a Source data file.

illustrated by the mixing lines in Fig. 2c, d. However, there is a significant positive correlation between C$_{37:4}$% C$_{38ME}$:C$_{38ET}$ in our MIS 6 samples when C$_{37:4}$% is larger than 4% (Pearson correlation, R = 0.92 and *p* < 0.01, Fig. 2b). This sense of correlation is consistent with C$_{37:4}$ alkenone variations due to varying proportions of alkenones produced by coccolithophores living in cold water (<10 °C) and those from warmer waters (-15–20 °C). Moreover, the Group 1 alkenone producers can generate double-bond positional isomers for C$_{37:3}$ and C$_{38:3}$[50], which are absent in our samples. This also suggests that the alkenones were not significantly sourced from Group 1 producers. The detailed

comparison of the abundance of alkenones (C$_{37}$–C$_{39}$) indicates that these alkenones were continually produced by Group 3 producers, coccolithophores.

## Polar water invading instead of stronger upwelling

Both lower growth temperature and high growth rate could stimulate the synthesis of C$_{37:4}$ by coccolithophores[39]. The upwelling is an important diver of modern coccolithophore productivity and sea surface temperature on the Iberian Margin[51]. So, the higher C$_{37:4}$% in the past could be explained as either the arrival of cold surface water

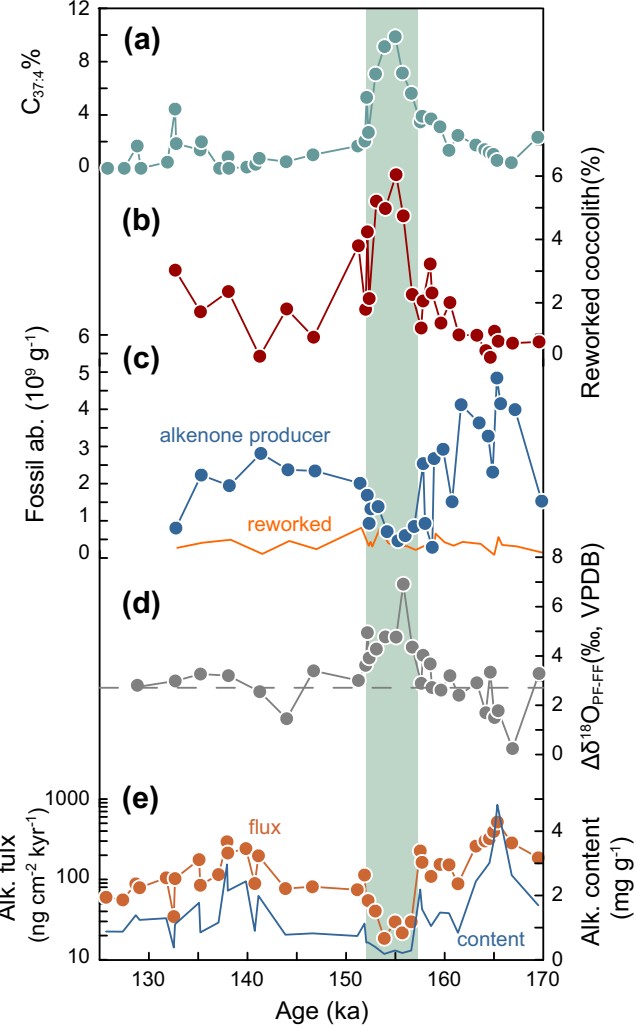

**Fig. 3 | Lower coccolithophore productivity during high tetra-unsaturated alkenones percentage ($C_{37:4}$%) period. a** $C_{37:4}$% in IODP U1385. The high $C_{37:4}$% period is highlighted by green dashing. **b** The percentage of reworked coccolith/nannofossils. **c** The abundance of fossils. **d** The oxygen isotope difference between planktonic foraminifera and fine fraction. The dashed line represents the mean value for all measurements. **e** Alkenone flux (orange) and alkenone content (blue). Note the axis of alkenone flux is in log10 scale. Source data of this figure are provided as a Source data file.

without an increase in coccolithophore growth rate or stronger upwelling with increased coccolithophore production. Reconstruction of coccolithophore productivity could distinguish between these alternatives.

Both nannofossils and molecular fossils suggest that the productivity decreased during the high $C_{37:4}$% periods. The coccolith abundance of alkenone producers decreased sharply during 165–150 ka, from ~5 × 10^9 g^{-1} to ~0.5 × 10^9 g^{-1}, coincident with the increase of $C_{37:4}$% (Fig. 3a, c). At the same time, the percentage of reworked coccolith increased from less than ~2% to 7% (Fig. 3b). The main reworked nannofossils were the typical Tertiary and late Cretaceous fossils (Supplementary Fig. S3a). In contrast with the percentage, the abundance of reworked coccolith remained stable through the MIS 6 (orange lines in Fig. 3c), which suggested that the increase in the percentage of reworked coccolith was not caused by the increase of supply of reworked carbonate, but by the decrease of coccolithophore productivity. Moreover, both alkenone content and flux support a decreased productivity during the interval of high $C_{37:4}$% (Fig. 3e). When the $C_{37:4}$% was higher than 4%, the

alkenone flux significantly decreased from ~150 ng cm^{-2} kyr^{-1} to only ~30 ng cm^{-2} kyr^{-1}.

Geochemical evidence also supports our interpretation. The $\delta^{18}O$ difference between planktic foraminifera, *Globigerina bulloides*, and the fine sediment fraction ($\Delta\delta^{18}O_{PF-FF}$) increased from 3‰ to maximum 7‰ during the higher $C_{37:4}$% period (Fig. 3d). This increase was mainly caused by the sharp decrease of fine fraction oxygen isotope ratios instead of large variations in foraminifera (Supplementary Fig. S4a). The increase of $\Delta\delta^{18}O_{PF-FF}$ in other Iberian Margin sites has been previously attributed to a significant increase in relative abundance of reworked fine carbonate[1]. The fine fraction carbonate is mainly composed by in situ coccoliths with similar oxygen isotope ratios as foraminifera, and reworked carbonate from sediments older than the Miocene featuring more negative $\delta^{18}O$ (Supplementary Fig. S3b). As part of reworked carbonate, the reworked fossils did not show significant increase in accumulation rate in our sediment. Thus, an increase of $\Delta\delta^{18}O_{PF-FF}$ at our site reflects a decrease of coccolithophore production, and thereby a smaller contribution of coccoliths to fine fraction.

Thus, diverse lines of evidence indicate that marine productivity sharply decreased during the cold events, consistent with previous assessments based on the *Florisphaera profunda* percentage, alkenone content and Cd/Ca ratio of foraminifera in this site and nearby sites[1,52,53]. We suggest that the higher $C_{37:4}$% on the Iberian Margin was controlled by the Polar water invading into the Iberian Margin, which led to lower surface temperatures and decreased coccolithophore productivities.

### Difference among alkenone thermometers on the Iberian Margin

Temperature estimations from alkenone unsaturation ratios on the Iberian Margin should be reliable due to the single source from Group 3 producers. SST reconstructions employing $U^{K}_{37}$, $U^{K'}_{37}$, and $U^{K}_{38ME}$, parallel each other and estimate similar absolute temperatures during the interglacial periods (Fig. 4). However, according to the applied core-top calibrations[27,54,55], the $U^{K'}_{37}$ temperature was ~2–4 °C warmer than the other two in the glacial period. Previous data-model comparisons of millennial temperature changes during MIS 3 suggested that the $U^{K'}_{37}$ suffered from warm biases towards summer temperature during glacial times[56], since couple ocean-atmosphere models of millennial abrupt coolings triggered by both freshwater hosing and self-oscillations of AMOC simulate larger cooling (-7–10 °C) during these stadials[57].

In our IODP U1385 records, the temperature reconstructions based on $U^{K}_{37}$ and $U^{K}_{38ME}$ feature larger amplitude stadial coolings, more similar to models[57], whereas all estimations given by $U^{K'}_{37}$ were warmer than simulations (Fig. 4 and Supplementary Fig. S2). Previously, Rosell-Melé[27] proposed that $U^{K}_{37}$ can better reflect the SST in high latitude marine sediment than $U^{K'}_{37}$ when the $C_{37:4}$% was higher than 5%. However, a new core top $U^{K}_{37}$ calibration from regions with only Group 3 alkenones may be required to confidently exploit the additional sensitivity of $U^{K}_{37}$ at cold temperatures, because the existing high latitude North Atlantic core top $U^{K}_{37}$ calibration may be affected by the significant contribution of Group 2i living under sea ice in some regions[20]. Furthermore, caution must be used when employing $U^{K}_{37}$ in high latitude marine sediments, because the alkenone, $C_{37:4}$, could be co-eluted with the alkenoate, $C_{36OET}$, resulting in a ~2–3% higher estimation of $C_{37:4}$% and cold bias of a few degrees in temperature reconstruction. This bias could be efficiently avoided by employing the gas chromatographic column, RTX-200[50], to fully separate the peaks of $C_{37:4}$ and $C_{36OET}$.

During some periods of our study, the sea surface temperature trends from alkenone thermometers differ from those estimated from Mg/Ca of *G. bulloides* (Fig. S5). This has been interpreted to result from the different seasonality of the alkenone and *G. bulloides* proxies[58,59]. Mg/Ca of *G. bulloides* records temperature during the narrow season window when water column is well-mixed[60]. In contrast, coccolithophores grow through the whole year on the Iberian Margin[61]

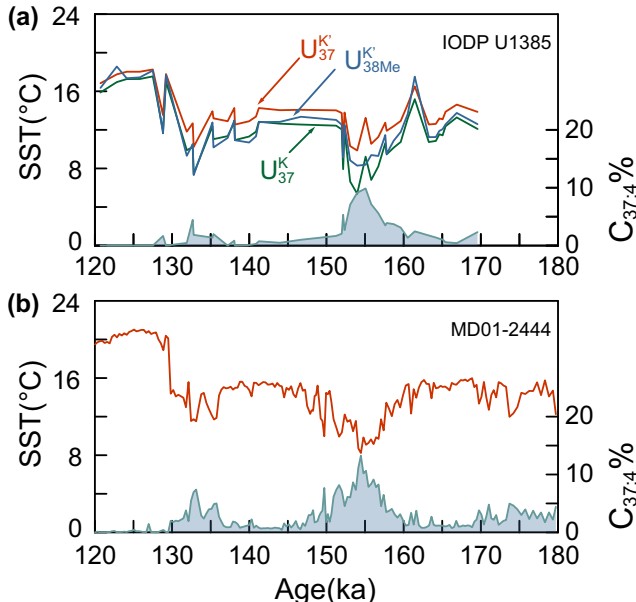

**Fig. 4 | Comparison of alkenone-based sea surface temperature (SST) on the Iberian Margin. a** Site IODP U1385. **b** Site MD01-2444. Red lines are SST reconstructed by $U_{37}^{K'}$ with Bayesian calibration[54]. Green lines are SST reconstructed by $U_{37}^{K}$ by Rosell-Melé[27]. Blue lines are SST reconstructed by $U_{38Me}^{K'}$ by Novak et al.[55]. The shaded areas are percentage of tetra-unsaturated alkenones ($C_{37:4}$%). Source data of this figure are provided as a Source data file.

and alkenone temperature is interpreted to integrate the mean annual temperature.

Taken as a whole, our results suggest that all the three alkenone thermometers can well reflect the timing of abrupt cooling events. The verification of a Group 3 origin of alkenones reopens the possibility to exploit the $U_{37}^{K}$ and $U_{38ME}^{K'}$ thermometers, which under robust calibrations may provide additional information about the amplitude of temperature changes during extreme cold periods.

### No freshening at the onset of cooling on the Iberian Margin

Because the Northern Hemisphere continental ice sheets store water of much lower $\delta^{18}O$ than the average surface ocean, the decrease of surface water $\delta^{18}O$ ($\delta^{18}O_{sw}$) has long been used as an indicator of meltwater-induced freshening of surface ocean[62-64]. In this work, the $\delta^{18}O_{sw}$ was reconstructed by the paired Mg/Ca ratio and $\delta^{18}O$ of *G. bulloides* (Fig. 5a, "Methods"). The $\delta^{18}O_{sw}$ during late MIS 6 was dynamic, varying between intermediate values, ~2‰ (VSMOW), and brief intervals of very positive values, ~4‰ which are similar to the Penultimate glacial maximum (Fig. 5a). The intermediate $\delta^{18}O_{sw}$ was consistent with simulations of intermediate ice volume during the same period[65]. From our comparison of $\delta^{18}O_{sw}$ and alkenone proxies on the same samples, no significant correlation can be found neither between $C_{37:4}$% and $\delta^{18}O_{sw}$ nor between SST and $\delta^{18}O_{sw}$ (Fig. 6a and Supplementary Fig. S6). In contrast, the $C_{37:4}$% is negatively correlated with alkenone SST (Fig. 6b–d). These comparisons provide further evidence that the $C_{37:4}$% should not be employed as a direct proxy for salinity on the Iberian Margin, but as a component recording cold temperatures in coccolithophore growth season.

For both the HE11 and MME, multiple stages of cooling can be identified (blue bars in Fig. 5), and these coolings were not always initiated with surface freshening. During the first stage of MME (MME-a in Fig. 5), with the onset of cooling, $\delta^{18}O_{sw}$ shifted to positive values indicating increased salinity. Only at the end of this cooling phase there was a negative shift of $\delta^{18}O_{sw}$ indicative of freshening (Fig. 5a–c). Similarly, in the second stage of MME (MME-b in Fig. 5), the onset of rapid

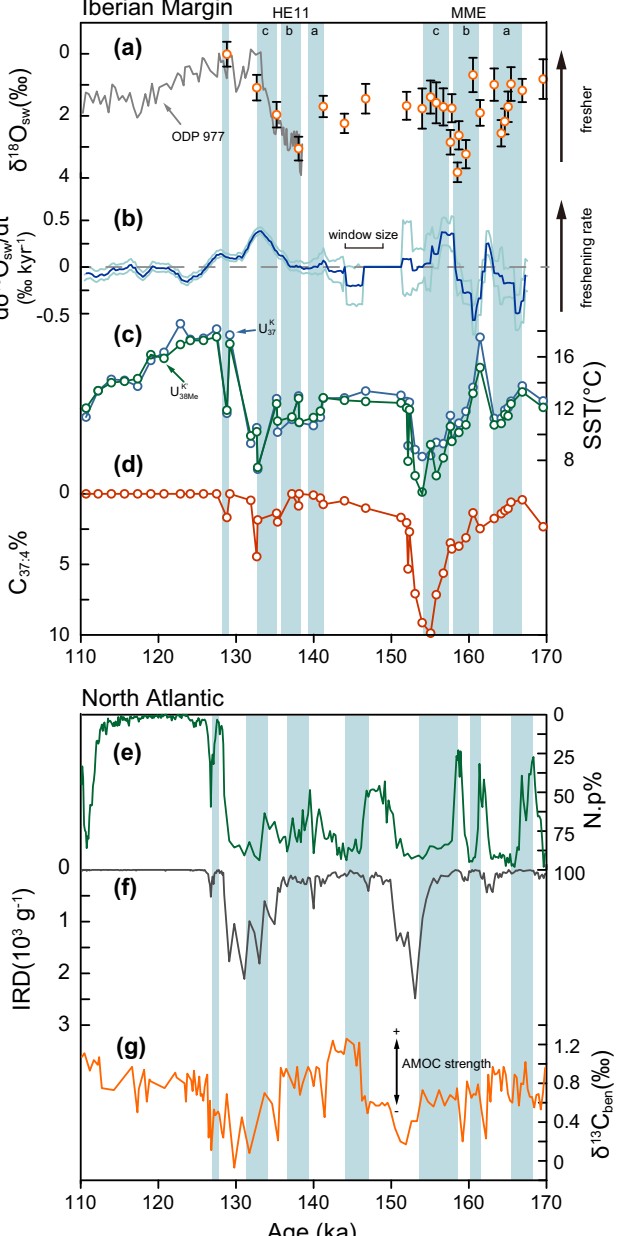

**Fig. 5 | The phase of cooling and freshening in abrupt climate events. a** Seawater oxygen isotope ratio reconstructed by paired Mg/Ca ratio and oxygen isotope of *Globigerina bulloides*. The orange dots are reconstructions in this work from IODP U1385. The gray lines are records in the ODP Site 977[104]. **b** The derivative of seawater oxygen isotope ratios shown in (**a**). The light blue lines represent the 95% confidence intervals in 10,000 Monte Carlo simulations considering the analytical errors of Mg/Ca and $\delta^{18}O_c$. Positive values represent larger freshening rates. **c** The sea surface temperature reconstructed by $U_{37}^{K}$ (green) and $U_{38Me}^{K'}$ (blue) in IODP U1385. **d** The $C_{37:4}$% in IODP Site U1385 (this work). **e**–**g** The percentage of *Neogloboquadrina pachyderma* (Np%), the ice-rafted debris (IRD) abundance and carbon isotope of benthic foraminifera in the ODP Site 983[8]. We highlight the cooling stages by alkenone temperature on the Iberian Margin and Np% in the North Atlantic with blue bars. Note that the HE11 is strictly defined as the period of elevated IRD which was not quantified in this core. Source data are provided as a Source data file.

cooling was accompanied by a positive shift of $\delta^{18}O_{sw}$. A freshening evident in the $\delta^{18}O_{sw}$ occurred later, at the end of MME-b event and was followed by the continued cooling through the last stage of MME (MME-c in Fig. 5). Though the data resolution is much lower, the first cooling

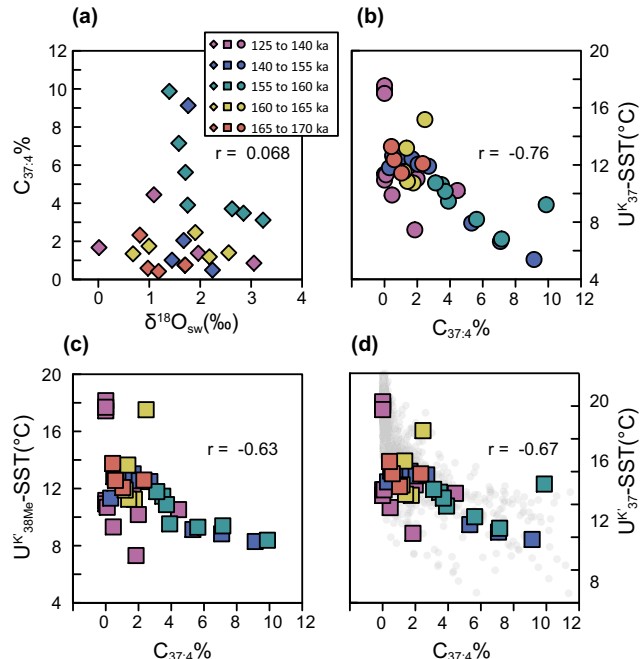

**Fig. 6 | Cross-plot of percentage of tetra-unsaturated alkenones ($C_{37:4}$%), $\delta^{18}O_{sw}$, and temperature in IODP U1385. a** No significant correlation between $C_{37:4}$% and $\delta^{18}O_{sw}$. **b–d** Negative correlations between $C_{37:4}$% and different temperature estimations. Spearman's rank correlation coefficients are shown in each panel. The gray dots in (d) are data from MD01-2444 in the past 424 kyr[4].

phase we resolve around HE11 (HE11-a in Fig. 5) was coincident with an increase in salinity indicated by a positive shift in $\delta^{18}O_{sw}$. The subsequent freshening of the termination (HE11-b, c in Fig. 5) was accompanied by a minor and then major cooling, as revealed in high-resolution speleothem records from Northwestern Iberia[66].

### Implication of abrupt climate events

The abrupt cooling events on the Iberian Margin were generally attributed to a reduction of AMOC intensity, entailing weakened oceanic heat transport and sea ice feedbacks leading to strong winter cooling[67–69]. A classical explanation of AMOC reduction focused on the freshening from meltwater in the North Atlantic initiating the AMOC decline[70–72]. However, our records reveal intriguing dynamics in the phasing of changes in $\delta^{18}O_{sw}$ and alkenone temperature. Due to age model uncertainties in the correlation of millennial events in marine sediments, the millennial coolings in our record cannot be independently correlated with precision to other marine records in the North Atlantic. Nonetheless, at site ODP 983 during this and other millennial events, the onset of IRD deposition has been shown to lag behind the surface cooling recorded by the percentage of *N. pachyderma*[8]. At least in some abrupt events, the freshening of the North Atlantic did not trigger the onset of cooling, but may have amplified the signal through a series of positive feedbacks during the cooling.

There are several explanations for the decoupling of freshening and cooling at the onset of abrupt climate events. The onset of cooling in the absence of regional freshening may be due to 'unforced' reductions of AMOC, which are not directly triggered by freshwater. These AMOC oscillations have been found in many coupled ocean-atmosphere-ice models[57, 73]. Alternatively, some models suggest that the Southern Ocean freshening, instead of North Atlantic, may also play important roles in the dynamics of AMOC[74–76].

In the middle of the MME event, we find that freshening began only after a pronounced cooling, consistent with a feedback mechanism

proposed by Marcott et al.[77]. In this mechanism, initial AMOC reduction leads to cooling and significantly expanded sea ice. Then, the growth of sea ice suppresses heat convection between surface and subsurface ocean and results in warming of the subsurface ocean under sea ice[77]. This subsurface warming is suggested to destabilize marine grounded ice sheets, which leads to larger freshwater fluxes and in favorable circumstances also the distal transport of significant IRD[78]. While most models of retreat of marine-based ice sheets due to subsurface warming have focused on the Hudson Bay system[77,79], it is proposed that during the MME, the IRD may originate from the Eurasian, rather the Hudson Bay, because no significant IRD can be found in the site IODP 1302/3 nor 1308 at this time[80]. Thus, based on our observations in the MME, similar behaviors of subsurface warming should be explored in models of marine-based sectors of the Eurasian Ice Sheet in the future.

Another diagnostic feature of MME is a weak bipolar seesaw effect. Margari et al.[42] observed a strong cooling on the Iberian Margin with a moderate warming in the Antarctic during the MME. This weak bipolar seesaw effect might be explained by the source of melting water, because the reduction of AMOC, which triggers the bipolar seesaw, could be very sensitive to the location of melting water injection[81]. While the carbon isotope of benthic foraminifera may serve as a general indicator of North Atlantic Deep Water depth or intensity (Fig. 5g), more quantitative proxies for the extent of AMOC reduction during the MME and HE11 are lacking. Some higher resolution Pa/Th records in the future would more fully elucidate this weak bipolar seesaw effect during the MME.

In summary, our detailed investigation of alkenone distribution from $C_{37}$ through $C_{39}$, confirms that alkenones on the Iberian Margin were consistently dominated by Group 3 producers over the 170 to 110 ka time intervals. Moreover, the periods of elevated $C_{37:4}$ abundance on the Iberian Margin should be interpreted as alkenone production generated in extremely cold conditions, but not directly a proxy for surface ocean freshening. We show that millennial events during the MME did not feature freshening as the initiation of abrupt cooling, providing important constraints on the role of freshwater forcing of AMOC variability during this time interval of large Eurasian and North American ice sheets and low glacial $CO_2$.

## Methods
### Sample selection
In total, 50 samples in the IODP Site 1385D and 1385E were selected for analyses. These samples covered the period from 170 ka to 110 ka in the MIS 6–5 based on the age model calibrated by tuning the benthic oxygen isotope to LR04[82]. Based on previous studies on alkenones[3,4], two significant $C_{37:4}$% peaks should be found during this period, the early one centered at -155 ka and the latter at -132 ka. The one around 155 ka witnessed the highest $C_{37:4}$% and longest duration of $C_{37:4}$% abnormal event in the last one million year[5] providing an ideal chance to test the source of alkenone and robustness of $C_{37:4}$% as a salinity proxy.

### Alkenone extraction and analyses
About 20–30 g sediment was first freeze-dried. Then, total lipids were extracted from the sediment using an Accelerated Solvent Extraction 350, with a dichloromethane and methanol mixture (5:1) at 100 °C. The hydrocarbon organic, ketone, and polar fractions were separated through silica gel columns and eluted with hexane, dichloromethane, and methanol, respectively. The ketone fraction, containing alkenones, was quantified on a Thermo Scientific Trace 1310 Gas Chromatograph (GC) coupled to a Flame Ionization Detector (FID) by injecting the sample in splitless mode. Chromatographic separation was achieved a Restek capillary column (105 m × 0.25 mm × 0.25 μm) RTX-200 ms, a 5-m guard column, and the following temperature program: 1 min at 50 °C, temperature gradient of 40 °C min$^{-1}$ to 200 °C, then to 300 °C at 3 °C min$^{-1}$, held at 300 °C for 40 min, and finally ramped to 320 °C at 10 °C min$^{-1}$ and held for 5 min. Peaks were

identified by comparing each peak's retention time with that of injected in-house alkenone standards.

Three alkenone-based thermometers are used in this study, $U^K_{37}$, $U^{K'}_{37}$, and $U^K_{38ME}$. The $U^K_{37}$ temperature is estimated by Rosell-Melé[27] calibrations, $U^K_{37} = -0.39 + 0.046\,T$ (°C). The $U^{K'}_{37}$ temperature is transferred by the BAYSPLINE[54]. The $U^K_{38ME}$ temperature is calculated via the new calibration by Novak et al.[55], $U^K_{38ME} = 0.016 + 0.032\,T$ (°C).

The alkenone content (AC) was quantified by peak areas in GC-FID. The relationship of peak area and absolute amount was calibrated by alkanes $C_{36}$–$C_{39}$ standards in five different concentrations ranging from 15 to 200 ng/μL. Then, the alkenone flux (AF) is calculated by

$$AF = \frac{AC}{\rho}SR \qquad (1)$$

where the $\rho$ is dry bulk density of sediment provided by IODP database (web.iodp.tamu.edu) and SR is the linear sedimentary rate calculated from age model. It should be noted that some alkenones had already been consumed during other measurements before quantifications. This could lead to 5% in maximum lower estimation for absolute amount, but the trend of alkenone content and flux should not be modified among samples.

### Stable isotope measurements

The raw sediments were filtered by 63 μm mesh to obtain the fine fraction. Based on light microscope observation, this fraction was mainly contributed by the carbonate particles smaller than 20 μm, and crushed foraminifera shells were rare. About 10 planktic foraminifera *Globigerina bulloides* (250–355 μm) were picked for oxygen and carbon isotope measurement. *G. bulloides* is selected since its calcification depth is estimated as ~50 m[83], and the finding of symbionts in *G. bulloides*[84] further supports a habitat within the euphotic zone. Then, the oxygen and carbon isotope of both foraminifera and fine fraction were measured by a Gas Bench II with an autosampler (CTC Analytics AG, Switzerland) coupled to ConFlow IV Interface and a Delta V Plus mass spectrometer (Thermo Fischer Scientific). The carbon isotope measurements were calibrated using international standards NBS 18 (−5.014‰), NBS 19 (+1.95‰) resulting in analytical errors smaller than 0.1‰ for both carbon and oxygen isotope ratios.

### Mg/Ca ratio analyses and $\delta^{18}O_{sw}$ estimations

Following previous studies estimating surface water $\delta^{18}O_{sw}$ on the Iberian Margin and in the North Atlantic[85–87], we use the paired Mg/Ca SST estimates and $\delta^{18}O_{calcite}$ from *G. bulloides* to characterize surface ocean freshening. On the Iberian margin, *G. bulloides* production occurs during periods of well-mixed and nutrient-rich water column (upwelling)[88]. In the North Atlantic, peak production of *G. bulloides* is closely coupled to the season of maximum chl-a concentration in surface waters, which occurs later in the year in colder higher latitude regions than in warmer subtropical zone[89]. In light of this ecology, the temperature recorded by Mg/Ca of *G. bulloides* tracks the temperature during its peak production season, a single season which may vary during different background states, and which may differ from the more integrated mean annual temperature signal interpreted from $U^{K'}_{37}$, as discussed for the Southern Iberian Margin[58]. In previous studies, the $\delta^{18}O_{sw}$ estimated by *G. bulloides* is comparable with that estimated by *Globigerinoides ruber*[90,91]. Thus, the $\delta^{18}O_{sw}$ estimated by *G. bulloides* can be used to trace the surface water freshening.

About 40 *G. bulloides* (250–355 μm) were picked for Mg/Ca ratio analyses. The shells were carefully crushed between two glass plates and then cleaned by protocols developed by Barker et al.[92] and Pena et al.[93]. Briefly, the cleaning was achieved by a first clay removal step; a reductive reagent attack to remove potential Mn–Fe oxide contaminant phases; an organic matter attack and a final weak acid leaching. After dissolved in 3 ml of ultra-pure 1% HNO₃, the samples

were analyzed on an inductively coupled plasma mass spectrometer (ICP-MS, PerkinElmer ELAN 6000) in the Scientific and Technological Centers of the University of Barcelona (CCiT-UB). An in-house high-purity standard solution was measured every four samples. All the data was subsequently corrected using a sample-standard bracketing method. The mean external reproducibility (2σ) for Mg/Ca was 4.00‰. The foraminifera calcification temperature was estimated by a sediment trap based calibration[94], which fully considered the in situ calcification temperature in the water column:

$$\frac{Mg}{Ca} = 0.955 e^{0.068T} \qquad (2)$$

Then, the seawater oxygen isotope ($\delta^{18}O_{sw}$) was estimated by the Mg/Ca-based temperature and foraminifera shell oxygen isotope ($\delta^{18}O_c$)[95,96] using the following equation:

$$T = 16.9 - 4.38\left(\delta^{18}O_c - \delta^{18}O_{sw}\right) + 0.1\left(\delta^{18}O_c - \delta^{18}O_{sw}\right)^2 \qquad (3)$$

where the $\delta^{18}O_c$ is in VPDB and $\delta^{18}O_{sw}$ is in VSMOW. The error of $\delta^{18}O_{sw}$ was quantified considering the analytical error of Mg/Ca and $\delta^{18}O_c$, as well as the regression error in Mg/Ca-temperature calibration via 10,000 times Monte Carlo simulations.

The derivative of seawater oxygen isotope ratios ($d\delta^{18}O_{sw}/dt$) was calculated using a running window method with a window size of 5 kyr and a window moving step of 0.25 kyr. A linear regression was preformed on $\delta^{18}O_{sw}$ data within the same window, and the slope of linear regression is the $d\delta^{18}O_{sw}/dt$ for this certain window. For windows contains less than 2 data points, $d\delta^{18}O_{sw}/dt$ of these periods are set as zero. The analytical errors of Mg/Ca and $\delta^{18}O_c$ were also considered in the $d\delta^{18}O_{sw}/dt$ calculations by using 10,000 times Monte Carlo simulations.

### Coccolith measurements

Coccolith assemblage counting and abundance estimation were performed on slides made by 'drop technique'[97]. At least 200 coccoliths and 10 fields of view were counted in a Zeiss ScopeA1 light microscope equipped with circular polarized light system for the coccolith abundance and accumulation rate analyses. The abundance of coccolith per gram sediment was calculated by the following equation:

$$Ab = \frac{N \times A}{f \times n \times w} \qquad (4)$$

where Ab is the abundance of coccolith (per g of sediment), $N$ is the total number of coccolith counted in the microscope, $A$ is the area of cover slip (24 mm × 24 mm), $f$ is the area of each field of view (0.0266 mm²), $n$ is the number of counted fields of view and $w$ is the weight of dry sediment on the cover slip. The identification of reworked nannofossils was based on the Nannotax3 website. Microscope examination revealed only very limited foraminiferal fragments in fine size fraction.

## Data availability

Source data of all new measurements are provided with this paper. Source data are provided with this paper.

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

## Acknowledgements

This study was supported by the Swiss National Science Foundation (Award 200021_182070 to H.S.) and ETH core funds to H.S. Y.H. acknowledges a sabbatical fellowship from Collegium Helveticum at ETH Zurich. We thank the Integrated Ocean Drilling Program (IODP) for providing the samples. The IODP is sponsored by the US National Science Foundation and participating countries under management of IODP Management International, Inc (IODP-MI). We thank Romain Alosius for assistance with sediment extraction and GC-FID, Madalina Jaggi for oxygen isotopic measurements, Ruigang Ma for helping in identifying reworked fossils, Lili Hu for picking foraminifera, Alexander J. Clark for help in ArcGIS, and Haowen Dang for suggestions in writing.

## Author contributions

H.Z. and H.S. designed this study. M.S. extracted the alkenone from bulk sediment. H.Z. and R.W. measured the alkenone with gas chromatograph with help of Y.H. H.Z. measured the oxygen isotope ratios and analyzed the coccolith. I.C. and J.T. analyzed the Mg/Ca ratio of foraminifera. B.W. contributed in providing worldwide surface alkenone data. OK contributed in explanation of background and deep discussions. H.Z. and H.S. wrote the manuscript with contributions from Y.H., R.W., I.C., J.T., M.S., O.K., and B.W.

## Competing interests

The authors declare no competing interests.
