## [Peer Review File · Nature Communications]

Surface ocean cooling led freshening on the Iberian Margin during the abrupt cooling events in Marine Isotope Stage 6REVIEWER COMMENTS

Reviewer #1 (Remarks to the Author):

This study provides an incisive look at the hitherto obscure paleoceanographic meaning of C37:4 in the high latitudes of the North Atlantic. It also discusses the relationship between freshwater runoff and cooling events in North Atlantic based on a reinterpreted alkenone proxy and the other multiple proxies. It is of high academic value in both organic geochemistry and paleoclimatology and has the potential for publication in Nature communications. Before the publication, more careful consideration should be given to some of the issues, which I referred to as general comments. In addition, several areas for improvement in the logic structure, including citations, which need to be corrected.

General Comments

According to the previous research, U1385 and MD01-2444 seem to have a long-term record of alkenone paleotemperature fluctuations, so why is this study focusing only on MIS 6? For example, I would like to know if the geochemical and paleoceanographic interpretations of this study can be applied to MIS 8 also including the huge peaks in C37:4%.

In this study, the number of coccoliths in the sediment and the $\Delta\delta^{18}\text{OPF-FF}$ are mainly used in discussing the productivity of the alkenone synthesizer. The former is potentially affected by changes in sedimentation rate and carbonate preservation, while the latter has uncertainties in the carbonate contamination in the FF. As an indicator of coccolithophore productivity, I think changes in alkenone fluxes should be incorporated into the discussion.

Although it may not be essential for the publication, I would like to suggest fossil-DNA analysis of haptophytes further to improve the reliability and novelty of this study. The authors have ruled out the existence of Group I and II based solely on alkenone composition, which has some high confidence. However, the characteristics of the alkenone patterns synthesized by Group I and II are still under study. Several Group II have been reported that synthesize alkenones similar to Group III (Nakamura et al., 2016, Organic Geochemistry). Would it be possible to perform DNA analysis on several critical stratigraphic levels of the studied cores to strengthen the interpretation of this study on the alkenone producers?

Specific comments

L48

“records in the Chinese Loess Plateau”

→What kind of records? Please clarify.

L59

“three groups of algae” is somewhat vague; how about “three groups of haptophyte” or “three clades in Isochrysidales”.

L79

The reference cited in #33 seems to be related to Group ii and not to *E. huxleyi*. Please recheck it.

L104

The period at the end of the sentence is missing.

L126-128

I can't catch the meaning of this sentence. The authors mention the correlation between C37:4% and C38Me:C38Et. Is it referring to the previously published culture experiments for Group III haptophytes? If so, an appropriate citation is required here.

L140-143

An explanation for "simulated millennial abrupt coolings⁵²" is lacking. What kind of experimentation is ref#52 based on? Please provide more specific evidence that the results of ref#52 can be used to assess the degree of cold events on the Iberian Margin during MIS3 and MIS6.

L173-175

Does the FF contain fragments of crushed foraminifera, etc.? If so, the interpretation of $\Delta\delta^{18}O$ may change.

L175-181

Reworked coccolith% and C37:4% behavior show similar trends. Could past alkenone producers have more C37:4 compared to the species at 150ka? Then, in that case, C37:4% increase may not necessarily indicate a change in water temperature or salinity.

L225-228

Hard to read. Please recapitulate the same and different aspects of the feedback mechanism described in ref#72 and newly suggested by this study.

L306

Please add how the measurement error in $\delta^{18}O_{sw}$ was calculated.

L314

What microscope did you use? Light microscopy or scanning electron microscopy?

L319 (Figure1)

U1385/2444 should be corrected to U1385/MD01-2444. The light blue lines are covered with the ocean, making it difficult to see.

L327(Figure 2)

In the manuscript, the existence of Group I is dismissed based on the absence of the C37:3 double peak. Therefore, (c) may not be necessary. I believe there should be no Group I contribution in this area. However, a straightforward interpretation of (c) would read as if Group I can present in a small amount.

L157

No journal information for Fujine et al., 2003. Please recheck for other deficiencies in the reference list.

Comments for Supplementary material

L36-37

It is difficult to discern whether the vertical axis SST in Fig S4(b) and (c) are based on Uk37 or Uk'38Me.

Reviewer #2 (Remarks to the Author):

Zhang et al. reported a detailed alkenone sea surface temperature record, along with other supporting proxy records, from the Iberian Margin during the Marine Isotope Stage 6. Their alkenone record show substantial cooling around ~155 ka and ~133 ka, coincident with the timing of the two strong IRD events reported in the North Atlantic. Their paired Mg/Ca and d18O measurements on *G. bulloides* do not suggest significant surface freshening at the time, especially at the onset of the two cooling events. They argue that increased alkenone C37:4 reflects colder temperature, rather than freshening, and thus that the temperature proxy UK37 is better than the other one UK'37 in reconstructing SST changes for those colder millennial events. They also conclude that during those millennial events, surface freshening did not always trigger the cooling.

The chain processes of millennial events are still intriguing. Although in model simulations, freshwater input is often used to trigger the weakening of AMOC and surface cooling, recent proxy studies suggest that these millennial events (IRD events) did not always start with surface freshening, and some studies event suggest they started with subsurface warming. This study adds to the growing body of evidence suggesting that freshwater input may not be the trigger. As the study would advance our understanding how abrupt climate change occurred, it should be of broad interest. I provide my comments below for authors' consideration during their revision.

1. I agree with authors that including C37:4 in the temperature index, i.e., the UK37, is better for those high latitude locations and/or glacial periods, but on the other hand, caution should be taken with the potential complexity by adding C37:4. Authors have found that UK'38ME is more consistent with UK37 rather than UK'37. However, physiologically, UK'38ME, which also does not include C38:4 in its definition, should behave similarly to UK'37 not UK37. Hence, it appears to me that the discrepancy in absolute temperature reconstructions perhaps results from calibration instead of its definition. For example, the authors used UK37 calibration by Rosell-Mele (1995), including core tops from high latitude North Atlantic, which is now known to be affected by Group 2i (Wang et al., 2021) or salinity.

Applying this calibration to authors' alkenone result, which authors claim to be solely from Group 3, could result from some uncertainty in SST reconstructions. Further, it seems that SST changes around ~155 ka reported by Martrat et al. (2007) are larger than those authors reported here, based on the same UK'37 index. Lastly, it is not an effective way using simulation results from ideal model experiments to justify proxy performance. Taking these uncertainties into consideration, authors could prefer to use UK37, but in authors' case, UK'37 performs reasonably well so that authors do not need to reject the use of UK'37.

2. It appears that authors accept that surface freshening was accompanied with the cooling events, perhaps not at the onset of the cooling, based on what is stated in title. This part is a bit confusing. Authors' d18Osw does not suggest particular surface freshening even at the time of max cooling, but authors appear to suggest that freshening occurred. I have checked authors Mg/Ca data, which do not show good correspondence to the alkenone SST results. Could authors use alkenone SST instead to derive d18Osw to see if it gets any better, assuming that the Mg/Ca proxy in this case is perhaps problematic? If authors indeed suggest that surface freshening was accompanied with the cooling events, then the occurrence of high C37:4 took place at both cooler and fresher conditions? Perhaps the two are indeed coupled during those millennial events, hence, back to my point #1, the potential complexity in C37:4 should be acknowledged (although I agree that C37:4 changes mainly reflect temperature in authors' case).

Wei et al. (2020) does not seem to be the correct ref. Please check!

Dear Reviewers,

First, we sincerely thank the two reviewers for your constructive comments. In this new version, we have implemented the main modifications as suggested:

- (1) Alkenone content and fluxes are carefully measured and plotted in Figure 3 as suggested by Reviewer#1;
- (2) In alkenone thermometers comparison section, we emphasize that the U_{37}^k still works well in detecting cooling events as suggested by Reviewer#2;
- (3) The methods of estimating seawater oxygen isotope ratios are better described in the Methods section as suggested by Reviewer#1;

Inspired by some of the questions posed by reviewers, we have sought to improve the clarity of the figure and text presentation to make the findings more accessible:

- (1) The order of Section 2.2 and Section 2.3 has been exchanged to make smoother logical progression of ideas;
- (2) The derivative of seawater oxygen isotope ratios is calculated to better illustrate the timing of freshening events (Figure 5);
- (3) A new figure, Figure 6, has been added into the main text to show that there was no correlation between $C_{37:4}\%$ and salinity (previously only shown in the supplementary).

The “point by point” response are listed in the following.

Reviewer #1

Q1: About sample selections. According to the previous research, U1385 and MD01-2444 seem to have a long-term record of alkenone paleotemperature fluctuations, so why is this study focusing only on MIS 6? For example, I would like to know if the geochemical and paleoceanographic interpretations of this study can be applied to MIS 8 also including the huge peaks in $C_{37:4}\%$.

Re: We thank the reviewer for prompting us to further clarify the motivation for this time period of focus. We have expanded the introduction to further detail that have chosen MIS 6 since it has a long duration significant $C_{37:4}\%$ peaks as well as IRD peaks, and represents intriguing question if the $C_{37:4}\%$ anomaly is due to strongly expanded sea ice, freshwater anomaly or cold. We also clarify that the choice of time period was motivated by the ongoing Paleoclimate Modelling Intercomparison Project (PMIP) protocol for the last deglaciation (Menviel et al., 2019) starting simulations at 140 ka. For this simulation, the sea ice coverage is an important boundary condition (e.g does the $C_{37:4}\%$ on Iberian Margin indicates sea ice coverage so far south?). The $C_{37:4}\%$ peaks at MIS 6 are comparable in magnitude to those characterizing MIS 8. We therefore expect that similar interpretations apply to other periods of anomalous $C_{37:4}\%$, and we outline an effective methodology for which the interpretation of the $C_{37:4}\%$ peak identified here, could be tested in other time intervals.

Q2: About alkenone fluxes. In this study, the number of coccoliths in the sediment and the $\Delta\delta^{18}O_{PF-FF}$ are mainly used in discussing the productivity of the alkenone synthesizer. The former is potentially affected by changes in sedimentation rate and carbonate preservation, while the latter has uncertainties in the carbonate contamination in the FF. As an indicator of coccolithophore productivity, I think changes in alkenone fluxes should be incorporated into the discussion.

Re: We thank the reviewer for this excellent suggestion to provide additional evidence for reconstructing coccolithophore productivity. We have added the alkenone contents and accumulation rates (fluxes) into revision. The methods of flux estimation are described in the Methods section and the new alkenone flux results have been provided Figure 4e. These results show low alkenone accumulation rates during the same intervals of low coccolith accumulation rates, and therefore strengthens our interpretation.

Q3: About DNA analyses. Although it may not be essential for the publication, I would like to suggest fossil-DNA analysis of haptophytes further to improve the reliability and novelty of this study. The authors have ruled out the existence of Group I and II based solely on alkenone composition, which has some high confidence. However, the characteristics of the alkenone patterns synthesized by Group I and II are still under study. Several Group II have been reported that synthesize alkenones similar to Group III (Nakamura et al., 2016, Organic Geochemistry). Would it be possible to perform DNA analysis on several critical stratigraphic levels of the studied cores to strengthen the interpretation of this study on the alkenone producers?

Re: We thank the reviewer for this suggestion. We agree with the reviewer on both points, that in future work paleo-DNA could be an exciting direction of study, but that it is beyond the scope of the current manuscript under consideration for publication. Our colleagues with expertise in paleo-DNA propose that chances of successful DNA extraction are much higher for more recent sediments. Therefore, a promising target might be the sediments with $C_{37:4}$ peaks during the last deglaciation, with ~120 kyr less aging and degradation than the MIS 6 interval of focus here. Projects targeting fossil-DNA from marine sediments also typically adopt special curation procedures which differ from standard handling of past IODP coring, such as shipboard freezing of the sediment immediately upon opening the core, which might be relevant to implement in future drilling campaigns on the Iberian Margin and elsewhere.

Specific comments

Line 48: “records in the Chinese Loess Plateau”. What kind of records? Please clarify.

Re: This sentence has been rewritten as:

“...loess grain size in the Chinese Loess Plateau”

Line 59: Three groups of XXX. “three groups of algae” is somewhat vague; how about “three groups of haptophyte” or “three clades in Isochrysidales”.

Re: We have change it as “three groups of haptophyte”

Line 79: Ref 33 The reference cited in #33 seems to be related to Group ii and not to E. huxleyi. Please recheck it.

Re: Yes, it should another paper from the same first author and same year. We have cited the correct one in this new version.

Line 104: The period at the end of the sentence is missing.

Re: Yes, fixed.

Line 126: I can't catch the meaning of this sentence. The authors mention the correlation between C_{37:4%} and C_{38Me:C38Et}. Is it referring to the previously published culture experiments for Group III haptophytes? If so, an appropriate citation is required here.

Re: We thank the reviewer for prompting us to clarify this sentence. Here the positive correlation between C_{37:4%} and C_{38Me:C38Et} is from our sediment samples. We use this positive correlation to exclude the potential mixings between Group 1-Group 3 and Group 2-Group 3. To make it clearer, this sentence has been changed to:

“However, there is a significant positive correlation between C_{37:4%} C_{38Me:C38Et} in our MIS 6 samples when C_{37:4%} is larger than 4% (Pearson correlation, R = 0.92 and p<0.01, Figure 2b).”

Line 140: An explanation for “simulated millennial abrupt coolings⁵²” is lacking. What kind of experimentation is ref#52 based on? Please provide more specific evidence that the results of ref#52 can be used to assess the degree of cold events on the Iberian Margin during MIS3 and MIS6.

Re: We thank both reviewers for prompting us to clarify this sentence. We have completely reorganized the section describing the various alkenone undersaturation indices and the previous model-data comparisons (Section 2.3). We now highlight that the inference of U^k₃₇ underestimating temperature amplitude is the conclusion of the previous model-data comparison which proposed summer bias during glacials as an explanation. We emphasize that confirmation of C_{37:4} origin from Group 3 haptophytes makes it possible to again consider the U^k₃₇ (including C_{37:4} in the index), as originally proposed by Rosell-Mele et al. (1998), and that this U^k₃₇ index has greater sensitivity to record cold temperatures and may contribute to resolving model-data discrepancies once robust calibrations are established.

Line 173: Does the FF contain fragments of crushed foraminifera, etc.? If so, the interpretation of Δδ¹⁸O may change.

Re: In the methods section we now state that in light microscope observations, there are only very limited fragments of foraminifera shells.

“Based on light microscope observation, this fraction was mainly contributed by the carbonate particles smaller than 20 μm, and crushed foraminifera shells were rare.”

Line 175: Reworked coccolith% and C_{37:4%} behavior show similar trends. Could past alkenone producers have more C_{37:4} compared to the species at 150ka? Then, in that case, C_{37:4%} increase may not necessarily indicate a change in water temperature or salinity.

Re: The reworked fossils were mainly from Miocene or older marine sediment outcrop on land. From such outcrop, due to preservation, especially considering the transport from land to ocean, we think the possibility of reworked alkenone is quite low. Because of the warm

temperature during the Miocene, if there were some contamination from ancient alkenone, we should find significant warm temperature bias. Also, there were no $C_{37:4}$ during the “green-house” period. The first occurrence of alkenone $C_{37:4}$ in the North Atlantic was ~1.3 million years ago (McClymont et al., 2008), with the North Hemisphere ice sheet extension. So, we suggest that the $C_{37:4}$ did not suffer from contaminations from reworked lipids.

Line 225: Hard to read. Please recapitulate the same and different aspects of the feedback mechanism described in ref#72 and newly suggested by this study.

Re: We thank the reviewer for the suggestion to clarify this sentence. We break this long sentence into three parts. Here we did not propose any new mechanism. Rather, we introduce the previous proposed theory and describe better how our new data fit with it.

“In the middle of the MME event, freshening began only after a pronounced cooling, consistent with a feedback mechanism proposed by Marcott, et al. ⁷². In this mechanism, initial AMOC reduction leads to cooling and significantly expanded sea ice. Then the growth of sea ice suppresses heat convection between surface and subsurface ocean and results in warming of the subsurface ocean ⁷².”

Line 306: Please add how the measurement error in $\delta^{18}O_{sw}$ was calculated.

Re: We use the Monto Carlo method to quantify the error of $\delta^{18}O_{sw}$, fully considering the analytical errors in Mg/Ca and foraminifera oxygen isotope ratios measurements, and the error is Mg/Ca-temperature regressions.

“The error of $\delta^{18}O_{sw}$ was quantified considering the analytical error of Mg/Ca and $\delta^{18}O_c$, as well as the regression error in Mg/Ca-temperature calibration via 10000 times Monte Carlo simulations.”

Line 314: What microscope did you use? Light microscopy or scanning electron microscopy?

Re: It is a light microscope. We add this information in line 315 now.

“At least 200 coccoliths and 10 fields of view were counted in a Zeiss ScopeA1 light microscope equipped with circular polarized light system for the coccolith abundance and accumulation rate analyses.”

Figure 1: U1385/2444 should be corrected to U1385/MD01-2444. The light blue lines are covered with the ocean, making it difficult to see.

Re: The light blue lines have been replaced by orange ones (as well as figure captions). And the full MD site name is now listed in panel a.

Figure 2: In the manuscript, the existence of Group I is dismissed based on the absence of the $C_{37:3}$ double peak. Therefore, (c) may not be necessary. I believe there should be no Group I contribution in this area. However, a straightforward interpretation of (c) would read as if Group I can present in a small amount.

Figure 1. Sites locations and climate records on the Iberian Margin. Please check the main text for figure captions.

Re: Yes, agree. We did not detect any C_{37:3} or C_{38:3} isomers. So, the Group 1 could be excluded. We have added a short clarification to our text:

“Moreover, the Group 1 alkenone producers can generate double bond positional isomers for C_{37:3} and C_{38:3}⁴⁹, which are absent in our samples. This also suggests that the alkenones were not significantly sourced from Group 1 producers.”

Beside the isomer evidence, we can also exclude the mixing between Group 1-Group 3 and Group 2-Group 3 by the slope of C_{37:4}% vs C_{38Me:Et}. If there is a mixing between Group 1-Group 3, the slope will be negative, but we observed a positive correlation between C_{37:4}% and C_{38Me:Et} in our Iberian Margin samples. This offers another important clue, so we kept the Figure 2c.

Moreover, all the Group I, II and III in this figure and its caption have been turned into Group 1, 2 and 3, to fit with main text.

Figure 2: No journal information for Fujine et al., 2003. Please recheck for other deficiencies in the reference list.

Re: Yes, it is not from a journal, but an IODP cruise report. We have given full details of this reference by adding the “Proceeding of Ocean Drilling Program, Scientific Results”.

Comments for Supplementary material

L36-37: It is difficult to discern whether the vertical axis SST in Fig S4(b) and (c) are based on Uk37 or Uk’38Me.

Re: We separate the Uk37 and Uk’38Me plots into two panels. Also, we have put part of this figure into main text Figure 6.

Figure S5 (previous Figure S4). Please check the supplementary for figure captions

Reviewer #2

Q1.1: I agree with authors that including C_{37:4} in the temperature index, i.e., the UK37, is better for those high latitude locations and/or glacial periods, but on the other hand, caution should be taken with the potential complexity by adding C_{37:4}. Authors have found that UK’38ME is more consistent with UK37 rather than UK’37. However, physiologically, UK’38ME, which also does not include C_{38:4} in its definition, should behave similarly to UK’37 not UK37. Hence, it appears to me that the discrepancy in absolute temperature reconstructions perhaps results from calibration instead of its definition. For example, the authors used UK37 calibration by Rosell-Mele (1995), including core tops from high latitude North Atlantic, which is now known to be affected by Group 2i (Wang et al., 2021) or salinity. Applying this calibration to authors’ alkenone result, which authors claim to be solely from Group 3, could result from some uncertainty in SST reconstructions.

Re: Yes, in theory, a C_{38Me}-thermometer with C_{38:4Me} should be more similar with the Uk₃₇. But the coccolithophore group 3 produced only limited C_{38:4Me} (see Figure 2 in Liao et al., 2022),

which on practical analytical level suggests it is currently not advantageous to include it in the index, as U_{38Me}^k is less influenced by $C_{38:4Me}$. Future studies from both lab culture and sediment measurements can test the influence of $C_{38:4Me}$.

We thank the reviewer for highlighting that the temperature bias derives largely from differing calibrations. Now we have reorganized this section and mention the calibration in Line 178-200.

“However, according to the applied core-top calibrations^{27,54,55}, the U_{37}^k temperature was ~2–4 °C warmer than the other two in the glacial period..... U_{37}^k and U_{38ME}^k thermometers, which under robust calibrations may provide additional information about the amplitude of temperature changes during extreme cold periods “

We also mention the potential influence of Group 2i on the previous calibration of U_{37}^k in line 152.

“As recommended by Rosell-Melé²⁷, U_{37}^k can better reflect the SST in high latitude marine sediment than $U_{37}^{k'}$ when the $C_{37:4\%}$ was higher than 5%. However, a new core top U_{37}^k calibration from regions with only Group 3 alkenones may be required to confidently exploit the additional sensitivity of U_{37}^k at cold temperatures, because the existing high latitude North Atlantic core top U_{37}^k calibration may be affected by the significant contribution of Group 2i living under sea ice in some regions²⁰”

Q1.2: Further, it seems that SST changes around ~155 ka reported by Martrat et al. (2007) are larger than those authors reported here, based on the same U_{37}^k index.

Re: We thank the reviewer for highlighting this detail. In addition to the MD01-2444 core published by Martrat et al. (2007), we also present but the U_{37}^k results from several other cores along the Iberian Margin. So, the structure and amplitude of events may be compared (new Figure S2 in supplementary). On the one hand, the U1385 record we present is lower in temporal resolution than the MD01-2444 record and thus may not have sampled the lowest temperature period which characterize the event, since our purpose was not to quantify the magnitude temperature change but rather to assess the origin of the $C_{37:4}$ peak. At the same time, however, among all of the U_{37}^k records there are subtle differences in structure before and after the 155 ka event. For example, the MD95-2042 record shows a local maximum in SST just prior to the 155 ka event, similar to our U1385 record, and the peak to 155 ka trough amplitude of this event in MD95-2042 is similar to that in U1385. While all published and our records reproduce the main features of the millennial cooling events, the cause of minor differences in structure and amplitude remains under discussion

Q1.3: Lastly, it is not an effective way using simulation results from ideal model experiments to justify proxy performance. Taking these uncertainties into consideration, authors could prefer to use U_{37}^k , but in authors' case, U_{37}^k performs reasonably well so that authors do not need to reject the use of U_{37}^k .

Re: We thank the reviewer for prompting us to clarify the purpose of the model-data comparison. We fully agree that model results alone are not sufficient criteria to favor one proxy over another. We have fully revised this section and retitled it as *“Differences among alkenone thermometers on the Iberian Margin”*. We highlight that the U_{37}^k , as well as U_{37}^k and U_{38ME}^k formulations resolve well the abrupt cooling events on the Iberian Margin.

We fully agree that model results alone are not sufficient criteria to favor one proxy over another. We now highlight that the inference of U_{37}^k , underestimating temperature amplitude is the conclusion of previous model-data comparison, which proposed summer bias during glacial as an explanation (Darfeuil et al., 2016).

We emphasize that confirmation of $C_{37:4}$ origin from Group 3 haptophytes makes it possible to again consider the U_{37}^k (including $C_{37:4}$ in the index), as original proposed by Rosell-Mele et al. (1998), and that this U_{37}^k index has greater sensitivity to record extreme cold temperatures and may contribute to resolving model-data discrepancies once robust calibrations are established.

Q2.1: About timing of freshening and cooling. It appears that authors accept that surface freshening was accompanied with the cooling events, perhaps not at the onset of the cooling, based on what is stated in title. This part is a bit confusing. Authors' $d^{18}O_{sw}$ does not suggest particular surface freshening even at the time of max cooling, but authors appear to suggest that freshening occurred.

Re: Our main conclusion is that the cooling is not always triggered by a preceding freshening. Yes, there are some freshening events during low temperature period. For example, a freshening happened during 157-152 ka (negative shifts in oxygen isotope and positive shifts in $d^{18}O_{sw}/dt$) during the cold period just before the lowest temperature recorded by U_{37}^k/U_{38Me}^k temperatures. We replot the Figure 5 by marking the direction of freshening and mentioning it in the figure caption. We also add a $d^{18}O_{sw}/dt$ plot to give a clear pattern of freshening.

Q2.2: Use alkenone temp. to calculate $\delta^{18}O_{sw}$. I have checked authors Mg/Ca data, which do not show good correspondence to the alkenone SST results. Could authors use alkenone SST instead to derive $d^{18}O_{sw}$ to see if it gets any better, assuming that the Mg/Ca proxy in this case is perhaps problematic?

Re: Standard practice in calculation of $\delta^{18}O_{sw}$ on the Iberian Margin and surrounding regions (Skinner and Shackleton, 2006; Peck et al., 2006; Tzedakis et al., 2018) has been the usage of foraminifera Mg/Ca paired to foraminiferal $d^{18}O$. Because the Mg/Ca from the same shells is optimally suited to provide the depth and seasonal temperature effect on ^{18}O during foraminiferal calcification.

It is important to use the correct temperature experienced by the foraminifera to avoid introducing artefacts in the calculation of $d^{18}O_{sw}$. Because based on modern observations, the water temperature is much more variable between depths and seasons compared with

Figure 5. The phase of cooling and freshening in abrupt climate events. Please check the main

oxygen isotope of water and salinity (<https://salinity.odysseallc.net/aq-climatology.htm>). Also, foraminifera living depth and season may not be identical to that of alkenone production and may therefore record somewhat different amplitude temperature evolution.

Analogously, if vital effects of coccoliths are well constrained, alkenone temperatures might be used with coccolith $\delta^{18}\text{O}$ to estimate the $\delta^{18}\text{O}_{\text{sw}}$ since they share the same production depth and season. However, in this specific site there are many reworked coccoliths and fine carbonate, which could render oxygen isotope signals in fine carbonate unreliable for this purpose.

Q2.3: C_{37:4} peaks during freshening. If authors indeed suggest that surface freshening was accompanied with the cooling events, then the occurrence of high C_{37:4} took place at both cooler and fresher conditions? Perhaps the two are indeed coupled during those millennial events, hence, back to my point #1, the potential complexity in C_{37:4} should be acknowledged (although I agree that C_{37:4} changes mainly reflect temperature in authors' case).

Re: We thank the reviewer for prompting us to clarify the timing of freshening.

We have provided an additional plot in the main text to show no relationship between C_{37:4}% vs $\delta^{18}\text{O}_{\text{sw}}$ and negative correlation between C_{37:4}% and cooling. (Figure 6). To make it easier to follow the time series records and see that fresher intervals do not have a unique association with cooling phases, we have added the $d\delta^{18}\text{O}_{\text{sw}}/dt$ into Figure 5. Figure 5 shows maxima in C_{37:4}% do not always coincide with minima in $\delta^{18}\text{O}_{\text{sw}}$ indicative of greatest freshening nor the highest $d\delta^{18}\text{O}_{\text{sw}}/dt$ indicative of fastest freshening.

Hence, we would say that cooling and freshening are decoupled considering the whole time sequence. However, in some specific conditions, the cooling and freshening may be coupled. **Only in the late stage** of MME and HE11 (not during the onsets of them), we can find higher C_{37:4}%, lower temperature and fresher seawater together. Because of the phasing, we believe that some previously described ocean-atmosphere-ice feedbacks explain the lead of cooling relative to freshening (see the Section 2.5). Based on lab culture works, the change of salinity did not directly modify the C_{37:4} abundance of a given strain, but low temperature does increase the abundance of C_{37:4} (Liao et al., 2022). So, we still suggest that the higher C_{37:4}% only reflects extreme cold temperature, though, in limited conditions, higher C_{37:4}% happened with fresher water.

We rephrase the whole Section 2.5 (Line 230-269) to explain this point better. For example, we have changed the sentence in Line 242 to emphasize that the decoupling of cooling and freshening occurred mainly at the beginning of abrupt climate events:

“There are several explanations for the decoupling of freshening and cooling at the onset of abrupt climate events.”

In this revision, we have emphasized further the potential complexity in C_{37:4}%, highlighting the potential productivity effect on C_{37:4}% based on culture studies, and emphasizing that C_{37:4}% should not be simplified as a simple temperature proxy, but a circumstance of extreme conditions:

“Moreover, the periods of elevated C_{37:4} abundance should be interpreted as alkenone production generated in extremely cold conditions...”

Ref: Wei et al. (2020) does not seem to be the correct ref. Please check!

Re: That is the correct reference. The journal Biogeosciences typed the U_{37}^k as $U_{37}^{k'}$ and we have corrected it to U_{37}^k .

Reference:

Darfeuil, S. et al. Sea surface temperature reconstructions over the last 70 kyr off Portugal: Biomarker data and regional modeling. *Paleoceanography* 31, 40-65, doi:10.1002/2015pa002831 (2016).

Martrat, Belen, et al. "Four climate cycles of recurring deep and surface water destabilizations on the Iberian margin." *Science* 317.5837 (2007): 502-507.

McClymont, Erin L., et al. "Expansion of subarctic water masses in the North Atlantic and Pacific oceans and implications for mid-Pleistocene ice sheet growth." *Paleoceanography* 23.4 (2008).

Menviel, Laurie, et al. "The penultimate deglaciation: protocol for Paleoclimate Modelling Intercomparison Project (PMIP) phase 4 transient numerical simulations between 140 and 127 ka, version 1.0." *Geoscientific Model Development* 12.8 (2019): 3649-3685.

Peck, Victoria Louise, et al. "High resolution evidence for linkages between NW European ice sheet instability and Atlantic Meridional Overturning Circulation." *Earth and Planetary Science Letters* 243.3-4 (2006): 476-488.

Rosell-Melé, Antoni. "Interhemispheric appraisal of the value of alkenone indices as temperature and salinity proxies in high-latitude locations." *Paleoceanography* 13.6 (1998): 694-703.

Skinner, L. C., and N. J. Shackleton. "Deconstructing Terminations I and II: revisiting the glacioeustatic paradigm based on deep-water temperature estimates." *Quaternary Science Reviews* 25.23-24 (2006): 3312-3321.

Tzedakis, P. C., et al. "Enhanced climate instability in the North Atlantic and southern Europe during the Last Interglacial." *Nature communications* 9.1 (2018): 4235.

REVIEWER COMMENTS

Reviewer #1 (Remarks to the Author):

The authors have responded constructively to most of the points I raised in my previous review, and I do not see any major additional requirements. The data quality and palaeoenvironmental interpretations for alkenone compositions are adequate. The climatological interpretations described in the revised chapter 2.5 are also understandable. However, I am less knowledgeable in this area and think that other experts' opinions should be fully reflected.

Reviewer #2 (Remarks to the Author):

Authors have made substantial effort in revising their manuscript and largely addressed my previous comments. In their response letter, they explained why they did not follow my suggestion using alkenone SST instead to derive $d18O_{sw}$, for which I accept authors' rationale. However, I feel that the issue is not fully resolved. It appears that authors interpret their bulloides Mg/Ca as surface temperature. The problem is then that the reconstructed sea surface temperature (SST) based on the alkenone and Mg/Ca proxies differs so dramatically. The particular warming around 159 ka in the Mg/Ca record is not seen in the alkenone records. Also, for the data point at 128.81 ka, the alkenone proxy (UK37) already registered interglacial temperature, while the Mg/Ca proxy still indicates very low SST. If authors remain their interpretation of bulloides Mg/Ca as surface temperature, then the discrepancy between the two proxies should be discussed at least. On the other hand, I'm wondering whether authors are willing to interpret the bulloides Mg/Ca as subsurface signal, as there exists some evidence suggesting subsurface warming preceding the max. surface cooling (i.e., Max et al., 2021, NC). However, in this case, then the $d18O_{sw}$, which is significantly affected by the temperature correction, may also reflect subsurface signal. Hence, the comparison between $\%C37:4$ and $d18O_{sw}$ will also be affected. Also intriguingly, $d18O$ of bulloides (i.e., without temperature correction) appears to follow the pattern of alkenone SST changes (I have made a couple of figures in authors' dataset, provided in the attachment). I do not have strong opinion which way to proceed, but wish that authors could refine the discussion in the relevant section.

Also, it is better to use K (capital case) in UK proxies.

Again, I believe that this study is of broad interest and could be considered by this journal.

Response to referees:

We thank the reviewer for reminding us to standardize the capitalization in the undersaturation index: all “U^K” have been corrected.

We also thank the reviewer for encouraging us to present clearer background on the production setting and seasonality of the proxy carriers (*G. bulloides* and alkenone producers), and to summarize previous consensus on the mechanisms for periods of divergent temperature SST trends. In this version, we have added discussions and explanations into **Section 2.3** and **Method 3.3** and **3.4**.

Habitat depth and niche of *G. bulloides*

In detail, in the methods section, we now provide more complete background information on the habitat depth and niche of *G. bulloides* and why it is used to indicate **surface** conditions, and provide further referencing of the use of *G. bulloides* to track surface (not subsurface) conditions in the North Atlantic, including surface ocean $\delta^{18}O_{sw}$. Previous studies seeking to evaluate **subsurface** conditions in the North Atlantic which employ deep calcifying species such as *G. truncatulinoides*, *G. inflata* and *N. pachyderma* (Farmer et al., 2011; Max et al., 2022), while *G. bulloides* is typically used in the North Atlantic to indicate surface conditions including the Iberian Margin (Farmer et al., 2011; Repschläger et al., 2015; Skinner and Elderfield, 2007).

In the new version, we note in Methods 3.3: “*Globigerina bulloides* (250–355 μm) were picked for oxygen and carbon isotope measurement. *G. bulloides* is selected since its calcification depth is estimated as ~50 meter (Ganssen and Kroon, 2000), and the finding of symbionts in *G. bulloides* (Bird et al., 2017) further supports a habitat within the euphotic zone.”

Also, in Methods 3.4: “ Following previous studies estimating surface water $\delta^{18}O_{sw}$ on the Iberian Margin and in the North Atlantic (Farmer et al., 2011; Repschläger et al., 2015; Thornalley et al., 2011), we use the paired Mg/Ca SST estimates and $\delta^{18}O_{calcite}$ from *G. bulloides* to characterize surface ocean freshening. On the Iberian margin, *G. bulloides* production occurs during periods of well-mixed and nutrient-rich water column (upwelling) (Salgueiro et al., 2008). In the North Atlantic, peak production of *G. bulloides* is closely coupled to the season of maximum chl-a concentration in surface waters, which occurs later in the year in colder higher latitude regions than in warmer subtropical zone. In light of this ecology, the temperature recorded by Mg/Ca of *G. bulloides* tracks the temperature during its peak production season, a single season which may vary during different background states, and which may differ from the more integrated mean annual temperature signal interpreted from U^{K}_{37} , as discussed for the Southern Iberian margin (Català et al., 2019). In previous studies, the $\delta^{18}O_{sw}$ estimated by *G. bulloides* is comparable with that estimated by *Globigerinoides ruber* (Anand et al., 2008; Elderfield and Ganssen, 2000). Thus, the $\delta^{18}O_{sw}$ estimated by *G. bulloides* can be used to trace the surface water freshening.”

The seasonality of *G. bulloides* on the Iberian margin and elsewhere in the North Atlantic does not affect the estimation of the surface ocean $\delta^{18}O_{sw}$, as long as the Mg/Ca SST and $\delta^{18}O_{calcite}$ are derived from the same proxy carrier as has been done in our and previous studies. Water temperature is much more variable between depths and seasons compared with oxygen isotope of water and salinity (<https://salinity.odyseallc.net/aq-climatology.htm>). Thus, we are confident that the $\delta^{18}O_{sw}$ reflects a surface meltwater signal which would be common across the growth period of *G. bulloides* as well as during the period of alkenone production. As we noted in the addition to methods section 3.4, previous work has carefully

compared the seawater oxygen isotope ratios reconstruction using *G. ruber* (much lower seasonality) and *G. bulloides* (Anand et al., 2008) and found them to yield very coherent trends over the last deglaciation. For this reason, *G. bulloides* has been widely used on the Iberian Margin and in the North Atlantic for detecting freshening events.

Alkenone thermometers vs Mg/Ca

Regarding the temperature histories, the difference between alkenone unsaturation indexes and Mg/Ca of *G. bulloides* has been well studied on the Iberian Margin (Català et al., 2019; Cisneros et al., 2016; Jiménez-Amat and Zahn, 2015). In the main text we briefly describe our and previous interpretations that contrasting proxy season is the main factor contributing to differences in time series trends of alkenone thermometers and Mg/Ca temperature estimates. Consistent with sediment trap, core top, and previous downcore studies, we propose that alkenone temperatures reflect mean annual temperature, whereas *G. bulloides* production is tightly linked to the upwelling season.

In discussion section 2.3, we now comment:

During some periods of our study, the sea surface temperature trends from alkenone thermometers differ from those estimated from Mg/Ca of *G. bulloides* (Figure S5). This has been interpreted to result from the different seasonality of the alkenone and *G. bulloides* proxies (Català et al., 2019; Jiménez-Amat and Zahn, 2015). Mg/Ca of *G. bulloides* records temperature during the narrow season window when water column is well mixed (Cisneros et al., 2016). In contrast, coccolithophores grow through the whole year on the Iberian Margin (Ausín et al., 2018) and alkenone temperature is interpreted to integrate the mean annual temperature.

We have plotted the Mg/Ca SST along with alkenone temperatures on supplementary figure S5.

Figure S4. Comparison between alkenone based temperature and Mg/Ca of *G. bulloides* on the Iberian Margin.

The deglacial amplitude of SST vs alkenones along the Iberian margin has been previously discussed as a consequence of evolving seasonality of peak *G. bulloides* production (Català et al., 2019). The temperature difference at ~159 ka is also likely a reflection of the contrasting proxy.

Reference

- Anand, P., Kroon, D., Singh, A.D., Ganeshram, R.S., Ganssen, G. and Elderfield, H. (2008) Coupled sea surface temperature–seawater $\delta^{18}\text{O}$ reconstructions in the Arabian Sea at the millennial scale for the last 35 ka. *Paleoceanography* 23.
- Ausín, B., Zúñiga, D., Flores, J.A., Cavaleiro, C., Froján, M., Villacieros-Robineau, N., Alonso-Pérez, F., Arbones, B., Santos, C., de la Granda, F., G. Castro, C., Abrantes, F., Eglinton, T.I. and Salgueiro, E. (2018) Spatial and temporal variability in coccolithophore abundance and distribution in the NW Iberian coastal upwelling system. *Biogeosciences* 15, 245-262.
- Bird, C., Darling, K.F., Russell, A.D., Davis, C.V., Fehrenbacher, J., Free, A., Wyman, M. and Ngwenya, B.T. (2017) Cyanobacterial endobionts within a major marine planktonic calcifier (*Globigerina bulloides*, Foraminifera) revealed by 16S rRNA metabarcoding. *Biogeosciences* 14, 901-920.
- Català, A., Cacho, I., Frigola, J., Pena, L.D. and Lirer, F. (2019) Holocene hydrography evolution in the Alboran Sea: a multi-record and multi-proxy comparison. *Climate of the Past* 15, 927-942.
- Cisneros, M., Cacho, I., Frigola, J., Canals, M., Masqué, P., Martrat, B., Casado, M., Grimalt, J.O., Pena, L.D. and Margaritelli, G. (2016) Sea surface temperature variability in the central-western Mediterranean Sea during the last 2700 years: a multi-proxy and multi-record approach. *Climate of the Past* 12, 849-869.
- Elderfield, H. and Ganssen, G. (2000) Past temperature and $\delta^{18}\text{O}$ of surface ocean waters inferred from foraminiferal Mg/Ca ratios. *Nature* 405, 442-445.
- Farmer, E.J., Chapman, M.R. and Andrews, J.E. (2011) Holocene temperature evolution of the subpolar North Atlantic recorded in the Mg/Ca ratios of surface and thermocline dwelling planktonic foraminifers. *Global and Planetary Change* 79, 234-243.
- Ganssen, G. and Kroon, D. (2000) The isotopic signature of planktonic foraminifera from NE Atlantic surface sediments: implications for the reconstruction of past oceanic conditions. *Journal of the Geological Society* 157, 693-699.
- Jiménez-Amat, P. and Zahn, R. (2015) Offset timing of climate oscillations during the last two glacial-interglacial transitions connected with large-scale freshwater perturbation. *Paleoceanography* 30, 768-788.
- Max, L., Nurnberg, D., Chiessi, C.M., Lenz, M.M. and Mulitza, S. (2022) Subsurface ocean warming preceded Heinrich Events. *Nat Commun* 13, 4217.
- Repschläger, J., Weinelt, M., Kinkel, H., Andersen, N., Garbe-Schönberg, D. and Schwab, C. (2015) Response of the subtropical North Atlantic surface hydrography on deglacial and Holocene AMOC changes. *Paleoceanography* 30, 456-476.
- Salgueiro, E., Voelker, A., Abrantes, F., Meggers, H., Pflaumann, U., Lončarić, N., González-Álvarez, R., Oliveira, P., Bartels-Jónsdóttir, H.B. and Moreno, J. (2008) Planktonic foraminifera from modern sediments reflect upwelling patterns off Iberia: Insights from a regional transfer function. *Marine Micropaleontology* 66, 135-164.
- Skinner, L. and Elderfield, H. (2007) Rapid fluctuations in the deep North Atlantic heat budget during the last glacial period. *Paleoceanography* 22.

Thornalley, D.J., Elderfield, H. and McCave, I.N. (2011) Reconstructing North Atlantic deglacial surface hydrography and its link to the Atlantic overturning circulation. *Global and Planetary Change* 79, 163-175.

REVIEWERS' COMMENTS

Reviewer #2 (Remarks to the Author):

Authors in this revision have sufficiently addressed the discrepancy between alkenone- and *G. bulloides* Mg/Ca-inferred sea surface temperature. I do not have further comments and recommend its acceptance.